# Contrastive Visual Data Augmentation

Yu Zhou [1] [*]   Bingxuan Li [1] [*]   Mohan Tang [1] [*]   Xiaomeng Jin [2]   Te-Lin Wu [1]   Kuan-Hao Huang [2] [3]
Heng Ji [2]   Kai-Wei Chang [1]   Nanyun Peng [1]

## Abstract

Large multimodal models (LMMs) often struggle to recognize novel concepts, as they rely on pre-trained knowledge and have limited ability to capture subtle visual details. Domain-specific knowledge gaps in training also make them prone to confusing visually similar, commonly misrepresented, or low-resource concepts. To help LMMs better align nuanced visual features with language, improving their ability to recognize and reason about novel or rare concepts, we propose a **Co**ntrastive visual **D**ata **A**ugmentation (**CoDA**) strategy. **CoDA** extracts key *contrastive* textual and visual features of target concepts against the known concepts they are misrecognized as, and then uses multimodal generative models to produce targeted synthetic data. Automatic filtering of extracted features and augmented images is implemented to guarantee their quality, as verified by human annotators. We show the effectiveness of **CoDA** on low-resource concept and diverse scene recognition datasets including iNaturalist and SUN. We additionally collect **NovelSpecies**, a benchmark dataset consisting of newly discovered animal species that are guaranteed to be unseen by LMMs. LLaVA-1.6 1-shot updating results on these three datasets show CoDA significantly improves SOTA visual data augmentation strategies by 12.3% (NovelSpecies), 5.1% (SUN), and 6.0% (iNat) absolute gains in accuracy. Code and data at contrastive-visual-data-augmentation.github.io

## 1. Introduction

Recent advancements in multimodal pre-training (OpenAI, 2023; Google, 2023; Hurst et al., 2024) and visual instruction tuning (Liu et al., 2023b;a; 2024c) have enabled impressive LMM abilities. However, as shown in Figure 1, it still

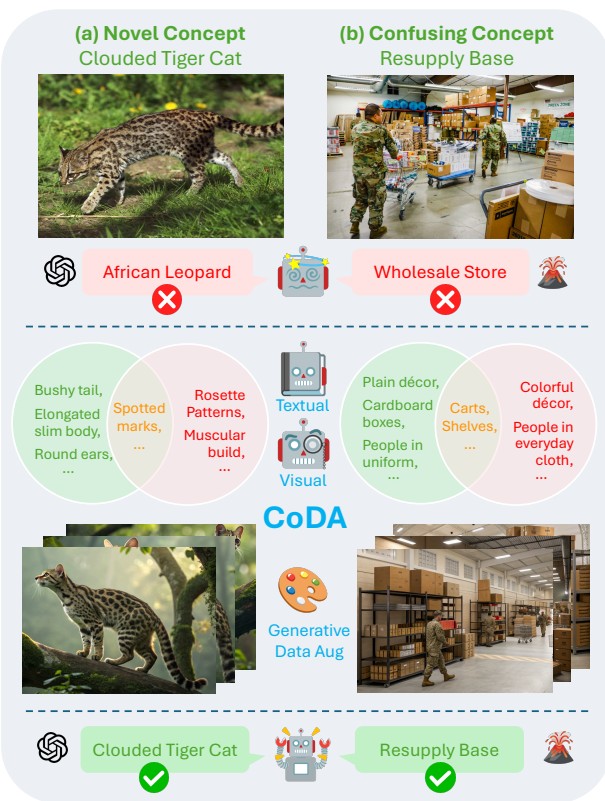

*Figure 1.* **CoDA** uses diffusion-generated synthetic data to help LMMs recognize novel and confusing concepts in the wild. The "Clouded Tiger Cat (L. pardinoides)" is a new animal species first described in April 2024, while "Resupply Base" is an example of a confusing concept for LMMs. Based on model failures (collected from GPT4o-2024-08-06 and LLaVA-NeXT 34B), **CoDA** extracts contrastive visual and textual features to generate synthetic image data for model updating.

remains a challenge for current state-of-the-art proprietary and open-source models to robustly recognize novel visual concepts (e.g."Clouded Tiger Cat" Figure 1a) and confusing / low-resource / commonly misrepresented visual concepts (e.g. "Resupply Base" Figure 1b).

In order to help models better acquire new visual concepts and distinguish confusable concepts, existing approaches straightforwardly 1). Fine-tune text decoder on new textual corpora to expand the concept base; and 2). Fine-tune both vision and text components on new web image-text pairs

---

*Equal contribution, interchangeable ordering. [1]UCLA [2]UIUC [3]TAMU. Correspondence to: Yu Zhou <yuzhou@cs.ucla.edu>.

*Proceedings of the 42nd International Conference on Machine Learning*, Vancouver, Canada. PMLR 267, 2025. Copyright 2025 by the author(s).

for visual concept acquisition. These approaches are ineffective due to data scarcity for certain concepts and data inefficiency caused by not knowing what precisely confused the models (3.1). As depicted in Figure 1, it can be difficult to obtain ample high-quality real images for novel concepts such as new animal species. While for confusing concepts, the problem usually lies with biased concept representation in web image-text data. For example: online images of "Resupply Base" mostly only consist of exterior views of the architecture without the interior details, which may cause the models to confuse it with a "Wholesale Store" that shares some interior features.

To help LMMs recognize and reason about novel and confusing concepts more robustly and efficiently, we propose **CoDA**, a **Co**ntrastive Visual **D**ata **A**ugmentation technique. For each target concept, **CoDA** first identifies a "confusable concept" that the LMMs finds most similar to the target. Then, it extracts contrastive textual and visual features of the target concept with respect to the confusable concept. The extracted features go through a filtering process based on discriminability and generability to make sure that: 1). The features are possessed by the target concept but not the confusable concept; and 2). The feature can be reliably generated by the text-to-image generative model and recognized by the LMMs. Afterwards, the features are passed to the text-to-image generative model to produce augmented visual instances of the target concept. To make sure that the features are indeed generated and recognizable by the LMMs, **CoDA** again uses the LMMs' zero-shot inference to rank and filter the augmented images. Finally, the resulting augmented images can be used to update the LMMs via low-rank adaptation, basic fine-tuning, in-context learning, or any other method of choice.

In addition to evaluating on existing datasets INaturalist and SUN, we create **NovelSpecies**, an annotated image dataset of newly discovered animal species in recent years. **NovelSpecies** allows the simple selection of species discovered after any model's latest knowledge cutoff date, ensuring the selected species were never seen by the model. Therefore, **NovelSpecies** is the perfect testbed for methods aimed at improving LMMs' novel concept recognition ability.

Comprehensive experiments with LLaVA-NeXT on the 3 datasets show **CoDA** performs surprisingly well in teaching LMMs novel and confusing concepts, significantly improving data efficiency compared to existing methods. In additional experiments, we show that **CoDA** is also able to improve novel concept recognition for traditional classifiers like ViT and proprietary LMMs such as GPT4o-mini. Finally, ablation experiments show that **CoDA** can be significantly improved by simply replacing its off-the-shelf components such as the text-to-image generation model with superior versions of similar models.

Our key contributions include:

- **CoDA**, a simple plug-and-play contrastive visual data augmentation method that can be used to effectively and efficiently improve LMMs' ability to recognize novel and confusing concepts. **CoDA** is also the first widely successful method using text-to-image generation for visual data augmentation.

- **NovelSpecies**, a new benchmark dataset of novel animal species discovered in recent years, providing an ideal benchmark for novel concept recognition. **NovelSpecies** currently consists of 2240 annotated images and will continue to be updated with future discoveries.

## 2. Related Works

Few-shot image recognition is a long-standing problem in the vision community. Early works in this area focused on improving traditional image classifiers on classifying existing concepts (Vinyals et al., 2016; Finn et al., 2017; Nichol, 2018; Dhillon et al., 2019; Tian et al., 2020; Bhagat et al., 2023; Afrasiyabi et al., 2022). On the other hand, while recent advancements in the training of vision language models (VLMs) and large multimodal models (LMMs) (OpenAI, 2023; Google, 2023; Hurst et al., 2024; Liu et al., 2023b;a; 2024c) have shown great promise and extensibility, they still severely lag behind traditional models in image classification, especially for low-resource, novel, and confusing concepts (Zhang et al., 2024; Cooper et al., 2024; Wu et al., 2023b; Yang et al., 2024; Ha et al., 2025).

While commonly used text-side VLM data augmentation strategies (Yuksekgonul et al., 2022; Yang et al., 2023; Liu et al., 2024e; Sharifzadeh et al., 2024) have little effect on this issue, a more promising technique to solve this is through visual data augmentation. This includes basic visual manipulations such as cropping, flipping, and rotation (Yang et al., 2022; Kumar et al., 2024); and more advanced model-based augmentation such as style transfer (Zheng et al., 2019; Chun & Park, 2021) and image mixing (Uddin et al., 2020; Xie et al., 2021; Hao et al., 2023). More recently, with the rise of controllable and promptable visual generative models, knowledge and feature editing-based augmentation methods (Liu et al., 2022; Wu et al., 2023a; Jin et al., 2024) have gained in popularity. Such methods generally focus on using multimodal data and general knowledge bases to guide image-editing models in creating augmented visual data based on existing images.

One main issue with current methods is that the augmented images they produce must be closely based on existing real images, which makes them unhelpful for novel concepts where real images are extremely rare, and mis-represented concepts where existing real images do not accurately depict the concept. Additionally, due to their close connection

to existing images, such augmented images usually lack visual frame structure and view variation. In contrast, our method **CoDA** can extract accurate and meaningful features from extremely limited multimodal data, and use text-to-image generative models to produce diverse high-quality augmented data for LMM updating.

## 3. Methods

As shown in Figure 2, **CoDA** consists of 4 major steps including contrastive textual and visual feature extraction, feature filtering, feature-controlled image generation, and augmented image filtering. Together these steps ensure **CoDA** reliably generates informative and high-quality augmented images that help LMMs recognize novel and confusing concepts.

### 3.1. Feature Extraction

**Textual Feature Extraction** In our exploratory experiments, we find that significant mis-recognition errors occur on low-resource or commonly mis-represented concepts in vision-language instruction fine-tuning and multimodal pre-training datasets, which the LMMs are trained on. For example, the LLaVA 1.6 (34B) model (Liu et al., 2024c), mainly tuned on LAION-GPT-4V(LAION, 2024) and ShareGPT-4V (Chen et al., 2023) datasets, has a strong tendency to mis-recognize interior images of "Resupply Base" as "Wholesale Store" (Figure 1). Unsurprisingly, we find that all related references of "Resupply Base" across the 3 instruction-tuning datasets only depict exterior views of the concept rather than interior views. While the concept itself is not a low-resource concept in existing text corpora, it is severely low-resource and also commonly mis-represented in vision-language instruction fine-tuning datasets.

To address this issue, we prompt LLMs to directly generate feature attributes of the target concept based on their existing knowledge, focusing on visual appearance, and avoiding hallucination for unfamiliar concepts. For this task, we use the cost-efficient GPT4o-mini model with chain of thought reasoning. Generally, textual feature extraction is most applicable for concepts that are high-resource in existing textual corpora, yet low-resource and/or commonly mis-represented in vision-language instruction-tuning and pre-training datasets. Here we do not try to classify which concepts fall under this criteria, but rather apply this step for all concepts. To ensure extracted feature quality and filter out hallucinated and/or non-visually-recognizable features, we pass all extracted features through an automatic filtering step, as described in 3.2.

We also considered other methods for textual feature extraction, including using knowledge bases (Jin et al., 2024), retrieval augmented generation, and LLMs with internet search. However, we believe currently the advantages brought by these methods do not out-weigh their complexity overhead, thus we opt for simplicity.

**Visual Feature Extraction** While textual feature extraction generally works well for pre-existing and non-hyper-domain-specific concepts that are prevalent in textual data sources, it tends to fail when either of the conditions are not met. For example, a large language model with a knowledge cutoff prior to June 2023 would not be able to provide meaningful features regarding the Apple Vision Pro device announced in July, or the new animal species "Clouded Tiger Cat (L. pardinoides)" first described by scientists in April 2024 (Figure 1). In addition to this weakness, LLM-based textual feature extraction is also unreliable when asked to provide detailed information regarding hyper-domain-specific concepts like the "Mazda MX-5 Miata RF" or the "Lear's Macaw (Anodorhynchus Leari)". In practice, we observe that for novel and hyper-domain-specific concepts, most of the LLM extracted textual features end up being filtered out by our automatic feature filtering module.

To address this weakness, we implement an additional visual feature extraction module based on VLMs. Given a single image of the target concept, the VLM is asked to extract its key visual features. When there is more than one image containing the target concept available, we use a LM to de-duplicate and summarize the combined extracted visual features from all images. For simplicity and cost-efficiency, we use the GPT4o-mini model for both visual feature extraction and feature de-duplication.

In contrast to textual feature extraction, visual feature extraction is most effective for hyper-domain-specific and novel concepts that are very rare or non-existent in textual corpora but have a limited number of visual examples. Thus, it well-complements textual feature extraction. Similarly, we do not attempt to classify which concepts fall under this criterion; instead, we apply this step to all concepts and rely on automatic filtering (3.2) to remove low-quality features.

**Contrastive Feature Extraction** While basic textual and visual feature extraction both aim to exhaustively list identifying features of the target concept, this is essentially an intractable task for complex concepts as it usually requires a huge number of features to fully describe them. For novel or low-resource concepts the LMM has likely never seen before, it is extremely difficult to teach the LLM the new concept using an incomplete description.

There are two potential solutions to this problem: (1). Leveraging hierarchical information to narrow down concept category and reduce descriptional features. (2). Illustrating the new concept based on contrastive differences from a similar existing concept the LMM already understands. Previous works in language and visual data augmentation (Jin et al., 2024) tend to use solution (1). However, its feasibility

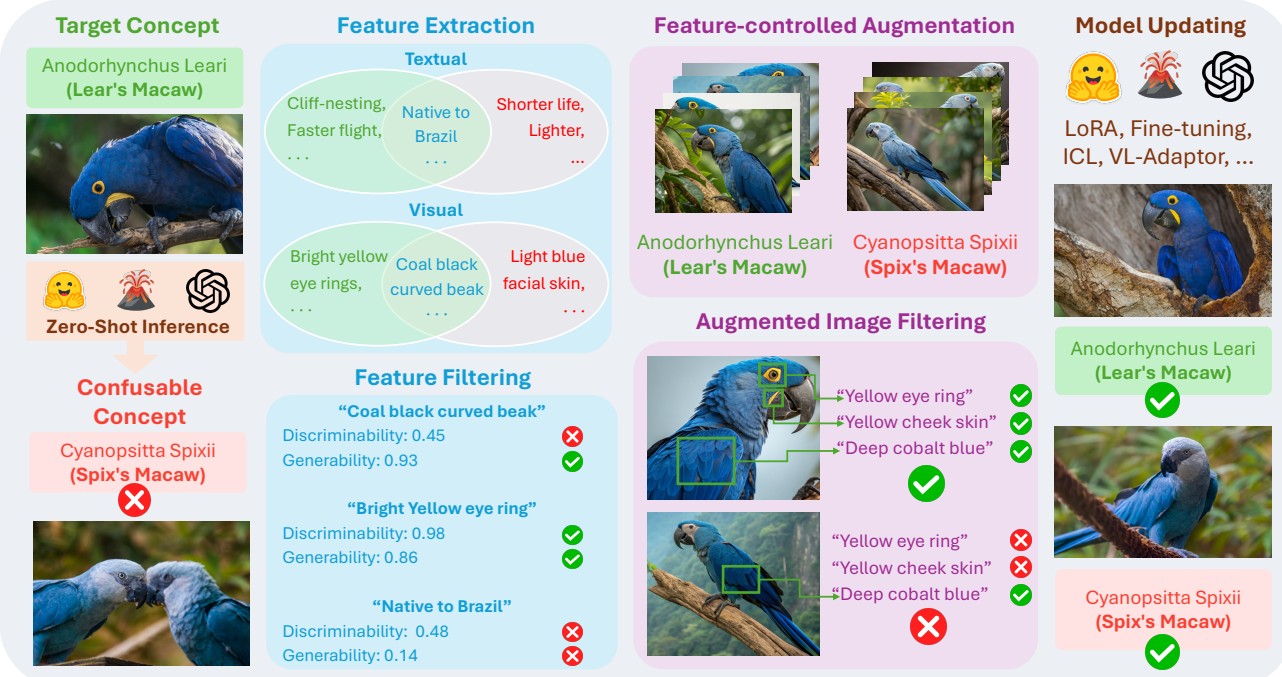

Figure 2. **The CoDA method**. Including Feature Extraction, Feature Filtering, Feature-controlled Augmentation, and Augmented Image Filtering. The target concept and misidentified concept are highlighted respectively. Specific feature filtering scores are for illustration only. Here the example concepts Anodorhynchus Leari (Lear's Macaw) and Cyanopsitta Spixii (Spix's Macaw) are from the iNaturalist (Van Horn et al., 2018) dataset, and augmented images are produced by the Recraft V3 model (Recraft.AI, 2024).

is contingent on the existence of a comprehensive textual knowledge base or tree-like structure that already includes the target concept. As discussed in Section 3.1, this is often not the case for novel concepts such as new electronic products (e.g. Apple Vision Pro) or new animal species (eg. Clouded Tiger Cat).

To enable the handling of novel concepts and remove the need for external databases, we adopt solution (2) and perform contrastive multimodal feature extraction for all target concepts. First, we use the LMM's zero-shot inference on the target concept $\mathcal{C}_{\mathcal{T}}$ to obtain the misidentified concept $\mathcal{C}_{\mathcal{M}}$. Then, we perform contrastive textual and visual feature extraction by querying LLMs and VLMs for visually identifying features that belong to $\mathcal{C}_{\mathcal{T}}$ but not $\mathcal{C}_{\mathcal{M}}$.

### 3.2. Feature Filtering

**Automatic Feature Filtering** After obtaining visually identifying features from contrastive textual and visual feature extraction, we filter them based on two key criteria:

1. **Discriminability** ($D(f, \mathcal{C}_T, \mathcal{C}_M)$) **:** measures whether a feature $f$ indeed differentiates the target class $\mathcal{C}_{\mathcal{T}}$ from the misidentified concept $\mathcal{C}_{\mathcal{M}}$ (check whether $f$ is a valid feature of $\mathcal{C}_{\mathcal{T}}$ but not $\mathcal{C}_{\mathcal{M}}$) .

2. **Generability** ($G(f, \mathcal{C}_T, \mathcal{C}_M)$) **:** measures whether a feature $f$ can be properly generated by the text-to-image generative model.

To calculate the Discriminability of a feature $f$ given the target concept $\mathcal{C}_{\mathcal{T}}$ and misidentified concept $\mathcal{C}_{\mathcal{M}}$, we compute the likelihood that CLIP (Radford et al., 2021) associates this feature with real images of the target concept compared to the likelihood that it is associated with real images of the misidentified class:

$$D(f, \mathcal{C}_T, \mathcal{C}_M) = \sum_{i \in I} \frac{\text{CLIP}(f, i_{\mathcal{C}_T}^{\text{real}})}{\text{CLIP}(f, i_{\mathcal{C}_T}^{\text{real}}) + \text{CLIP}(f, i_{\mathcal{C}_M}^{\text{real}})}$$

Here we use an equal number of images of the target and misidentified concepts. A score below 0.5 indicates that the feature is more likely to be associated with the misidentified class rather than the target class. To ensure that selected features are more strongly associated with the target class, we filter out all features with Discriminability below 0.6. This method avoids the CLIP score bias against smaller features by only comparing feature association with the two classes and not relying on the absolute CLIP score.

Generability is calculated for all features that pass the Discriminability threshold. We prompt the T2I generative model $g$ to generate synthetic images of the target concept that contains the feature $f$, and then compare the average CLIP similarity between $f$ and those generated images against the average CLIP similarity between $f$ and $i_{\mathcal{C}_M}^{\text{real}}$:

$$G(f, \mathcal{C}_T, \mathcal{C}_M, g) = \sum_{i \in I} \frac{\text{CLIP}(f, i_{\mathcal{C}_T}^{\text{synthetic}})}{\text{CLIP}(f, i_{\mathcal{C}_T}^{\text{synthetic}}) + \text{CLIP}(f, i_{\mathcal{C}_M}^{\text{real}})}$$

Here we rank all remaining features by their Generability score and select the top 5 features to be passed to the text-to-image generative model (as current diffusion models usually have limited text encoder attention span). This step identifies features that not only help distinguish the target concept, but also can be effectively rendered by the text-to-image generative model in synthetic images, which is critical to the success of synthetic data augmentation.

Our automatic feature filtering module based on Discriminability and Generability ensures feature quality and limits the information loss between features and the generated augmented images. The remaining features are used for image generation and improving in-context recognition ability in inference prompts. We further verify the quality of remaining features with human evaluation in Sec.3.4.

### 3.3. Image Generation and Verification

**Image Generation**    After feature extraction and filtering based on Discriminability and Generability, we pass the selected features to a text-to-image generative model to generate augmented visual data. We experiment with both SOTA open-weights (Esser et al., 2024; Stability AI, 2024) and proprietary (Recraft.AI, 2024) models.

**Verification**    To ensure final images for augmentation contain our extracted and filtered target concept features, we propose a simple automatic verification metric that checks whether desired features are recognized in the augmented images by the LMM we want to update: Given the vanilla LMM $\mathcal{M}$, a set of features $\mathcal{F}$, and an augmented image $i^{\text{synthetic}}$, the feature satisfaction rate $S(i^{\text{synthetic}}, F, M)$ for each augmented image:

$$S(i^{\text{synthetic}}, \mathcal{F}, \mathcal{M}) = \frac{\sum_{f \in \mathcal{F}} \mathbf{1}\{\mathcal{M}(f, i^{\text{synthetic}})\}}{|\mathcal{F}|}$$

Here $\mathcal{M}(f, i^{\text{synthetic}})\}$ returns true if the feature $f$ is recognized in the image $i^{\text{synthetic}}$. Afterwards, we filter out all images with $S(i^{\text{synthetic}}, F, M) < 1.0$, keeping only augmented images that fully match all target concept features.

### 3.4. Human Evaluation

| Image Type | Target Concept (%) | Misidentified Concept (%) | Inter-Annotator Agreement ($\kappa$) |
|---|---|---|---|
| Real | 92.51 | 14.32 | 0.87 |
| Synthetic | 83.97 | - | 0.82 |

*Table 1.* **Human eval of extracted features and augmented images.** 3 external annotators are asked to answer (yes/no) to whether the extracted and filtered features are present in the corresponding real and synthetic images. IAA based on Fleiss' Kappa.

To verify the reliability of our feature filtering and augmented image verification modules, we conduct human evaluation on a subset of iNaturalist and the novel animal species dataset. For target concepts, we select 100 image-feature pairs for both real and augmented synthetic images. We also select 100 image-feature pairs for real images of misidentified concepts. 3 external human annotators are asked to label whether they believe the given feature belongs to the concept in the corresponding image.

Results in Table 1 show human annotators overwhelmingly agree that the final extracted features belong to the target concept (92%) but not the misidentified concept (14%). The augmented synthetic images of the target concept also likely contains the desired features (83%), though as expected, there is some information loss between the text-to-image generation step. In addition, the three independent annotators generally agreed in their response (>0.8 IAA).

### 3.5. In-Context Inference for Enhanced Recognition

In addition to updating the LMM with augmented data, we can further boost performance by integrating the extracted features into the inference prompt. For each query, we can append a concise list of the most discriminative and generable features of the target and confusable classes. These features serve as an in-context guide, focusing the LMM's attention on critical distinguishing attributes. By explicitly highlighting what to look for (and what not to mistake it for), the model more reliably identifies the correct concept.

## 4. NovelSpecies Dataset

Proprietary LMMs like GPT4o (Hurst et al., 2024) and Gemini (Google, 2023) are trained on vast online text-image data and proprietary data, both non-public and impossible to inspect. Some open-source and open-data LMMs such as LLaVA (Liu et al., 2024b;d) are trained on publicly available image-text datasets. However, the text encoders used by such models are often not open-data, for example LLaVA-1.6 34B uses the closed-data Yi-34B model as its language backbone. Even in the rare cases where both image-text training data and text encoder training data are publicly available, it is still difficult to ascertain whether concepts in your benchmark were seen by your LMM through indirect data leakage (i.e. partial / paraphrased mentions). Due to the above issues, it is difficult to evaluate true novel concept recognition ability with existing datasets.

One way to bypass this problem with 100% guaranteed success is to use a dataset that only contains concepts created / discovered after the LMM's knowledge cutoff, i.e. the latest knowledge cutoff date among all of its textual / visual sub-components. Based on this idea, we curate **NovelSpecies**, a dataset of novel animal species discovered in each recent

year, starting with 2023 and 2024. We provide detailed information for each species, including time of discovery, latin name, common name, family category, textual description, and more. Data will be released upon publication.

To create **NovelSpecies**, we start by collecting the list of species first described in each year by Wikidata (Wikidata, 2024). Then, to make sure we can curate a visual benchmark of novel species, we manually annotate and filter out extinct species and species with too few publicly available images. After filtering, we end up with a dataset of 64 new species, each consisting of 35 human-verified image instances, thus a total of 2240 images. The images are split into training, validation, and test sets. For each species, there are 5 training images, 15 validation images, and 15 test images. This data split is consistent with our goal of creating a benchmark dataset for novel concept recognition, where the maximum number of training instances for a completely unseen concept can range from 1 to 5.

## 5. Experiments

### 5.1. Datasets and Baselines

To evaluate **CoDA**'s ability to improve novel and confusing concept recognition in LMMs, we experimented with **CoDA** and other relevant baselines on three different datasets:

1. **The iNaturalist Dataset** (Van Horn et al., 2018) is a challenging natural world concept recognition benchmark for LMMs due to its extensive highly domain-specific and fine-grained species categories and inclusion of rare and low-resource species classes.

2. **The SUN Dataset** (Xiao et al., 2010) is a widely used large-scale scene recognition dataset that contains rich and confusing visual scenes. Correctly recognizing the scenes requires fine-grained visual reasoning and understanding of the scenes.

3. **NovelSpecies Dataset** (Sec.4) is our new dataset consisting only of novel animal species concepts that LMMs are guaranteed to have never encountered in their training or fine-tuning.

For each dataset, we use an automatic data selection strategy A.2 to find a subset of challenging concepts that the model fails to recognize. Then, we apply **CoDA** along with 3 other visual data augmentation baselines:

1. **All Real** uses an all real augmented image set. In the Fixed Real Data setting, this means using the 5 real images provided. In the Fixed Compute setting, this means using unlimited real images to match the total number of real + synthetic images in other settings.

2. **Cropping and Flipping** are widely used traditional visual data augmentation strategies. We include them here for direct comparison with **CoDA** and other existing feature-based augmentation methods.

3. **ARMADA (Jin et al., 2024)** is the current state-of-the-art feature-based visual data augmentation strategy for concept recognition and image classification.

In addition to these 3 baselines, we also include ablations of **CoDA** with non-contrastive textual and visual features, i.e. w/o contrastive guidance from confusable concepts (3.1) nor discriminability-based feature filtering (3.2).

### 5.2. Main Experiment

For our main experiment, we consider two different resource settings that correspond to common real-world scenarios:

**Fixed Real Data**     Under the fixed real data setting, we only have access to 5 real images for each concept. Each data augmentation strategy may generate 1-5 synthetic images. Then, the model is LoRA-adapted on the combined real and synthetic images. This setting simulates real-world scenarios, where there isn't sufficient real training data for certain concepts. This is common for novel concepts, hyper-domain-specific concepts, and long-tail distributed datasets. In these scenarios, the quality and effectiveness of synthetic augmented data is especially instrumental to the updated model's performance.

Experiment results across the 3 datasets show that **CoDA** consistently outperforms existing traditional and feature-based data augmentation methods in the Fixed Real Data setting. When augmenting the training set with just a single synthetic image, **CoDA** is able to achieve 11.8% (NovelSpecies), 10.0% (SUN), and 17.8% (iNat) absolute gains in accuracy compared to using all real images. It further outperforms the best existing baseline augmentation methods by 5-12% absolute gains. We also observe that the ablated performance of **CoDA** (w/o contrastive) is still significantly above traditional and image-editing-based augmentation baselines while being almost consistently below **CoDA**'s performance. This shows the benefits of text-to-image generative augmentation methods compared to existing methods, as well as the benefits of fine-grained textual features during inference. This also highlights the need for contrastive feature selection and discriminability-based feature filtering. We find that increasing the number of augmented synthetic images does not necessarily improve updated model performance; this may be attributed to the fact that all generated images are ranked and selected from the same pool, with the first image being of the highest quality. Finally, the largest improvement over existing baselines can be seen in **NovelSpecies**, where **CoDA** methods involving visual features achieve the highest performance. This makes sense as the visual feature extraction method is designed to be robust to novel concepts with little textual documentation.

| Dataset | Augmentation Method | Feature Type | Fixed Real Data (Real:Syn) | | | | Fixed Compute (Real:Syn) | | |
|---|---|---|---|---|---|---|---|---|---|
| | | | 5:0 | 5:1 | 5:3 | 5:5 | 20:0 | 10:10 | 0:20 |
| NovelSpecies (Sec.4) | Baselines | All Real | **61.2** | - | - | - | - | - | - |
| | | Cropping | - | 60.4 | 60.4 | 59.5 | - | - | - |
| | | Flipping | - | 60.7 | 62.9 | 60.1 | - | - | - |
| | | ARMADA | - | 60.7 | 60.2 | 61.2 | - | - | - |
| | CoDA (w/o contrastive) | Textual | - | 69.1 | 68.6 | 70.5 | - | - | - |
| | | Visual | - | 71.8 | 72.6 | 71.7 | - | - | - |
| | | T+V | - | 70.3 | 65.1 | 70.1 | - | - | - |
| | CoDA | Textual | - | 72.0 | 69.2 | 70.3 | - | - | - |
| | | Visual | - | **73.0** | **72.8** | 71.8 | - | - | - |
| | | T+V | - | 70.1 | 72.6 | **73.0** | - | - | - |
| SUN (Xiao et al., 2010) | Baselines | All Real | **73.4** | - | - | - | **74.3** | - | - |
| | | Cropping | - | 78.3 | 75.8 | 76.3 | - | 77.3 | 76.4 |
| | | Flipping | - | 75.7 | 78.4 | 74.8 | - | 75.2 | 76.1 |
| | | ARMADA | - | 75.9 | 78.3 | 77.6 | - | 76.2 | 76.8 |
| | CoDA (w/o contrastive) | Textual | - | 80.6 | 79.7 | 79.4 | - | 81.3 | 80.8 |
| | | Visual | - | 81.3 | 81.6 | 79.3 | - | 80.0 | 80.8 |
| | | T+V | - | 82.7 | 80.7 | 80.4 | - | 82.8 | 82.1 |
| | CoDA | Textual | - | 79.2 | **83.2** | 82.3 | - | 82.8 | 82.1 |
| | | Visual | - | 82.3 | 81.7 | 82.2 | - | 81.8 | **83.1** |
| | | T+V | - | **83.4** | 81.7 | **82.6** | - | **83.3** | 82.1 |
| iNaturalist (Van Horn et al., 2018) | Baselines | All Real | **49.2** | - | - | - | **64.3** | - | - |
| | | Cropping | - | 59.7 | 58.8 | 62.2 | - | 61.4 | 63.9 |
| | | Flipping | - | 61.0 | 61.1 | 62.3 | - | 62.1 | 62.7 |
| | | ARMADA | - | 60.1 | 60.7 | 61.1 | - | 61.6 | 58.5 |
| | CoDA (w/o contrastive) | Textual | - | 63.9 | 64.6 | **66.5** | - | 65.6 | 63.2 |
| | | Visual | - | 65.0 | 64.7 | 64.3 | - | 65.6 | 63.2 |
| | | T+V | - | 62.8 | 64.4 | 62.3 | - | 64.4 | 63.4 |
| | CoDA | Textual | - | 63.9 | **67.8** | 62.6 | - | 65.0 | **64.9** |
| | | Visual | - | **67.0** | 66.0 | 65.1 | - | 62.5 | 60.9 |
| | | T+V | - | 63.5 | 65.0 | 64.6 | - | **67.0** | 64.1 |

*Table 2.* **Main experiment results on INaturalist, SUN, and NovelSpecies under Fixed Real Data and Fixed Compute settings:** Experiments are defined by the number of Real:Synthetic images used. For example, 5:1 means the model uses 5 real images and 1 synthetic image for each concept class at training time. All results are in terms of LLaVA-1.6 34B concept recognition accuracy (%). Best performance scores for each setting and scores using all real data are highlighted in **Red** and **Green**, respectively.

**Fixed Compute** Under the fixed compute setting, we assume access to unlimited real and synthetic images. However, the fine-tuning budget can only support a total of 20 images, allowing different percentages of real and synthetic images, from 0% synthetic (20:0) to 100% synthetic (0:20). This setting simulates real-world scenarios, where there is abundant real data. In such cases, the question is whether to just use all real data to update the model, or to include a non-trivial amount of augmented synthetic data. Traditionally, real data is always preferred due to perceived higher-quality. However, **CoDA**'s effectiveness in the Fixed Real Data setting prompts us to test the possibility of it being beneficial to include synthetic data even when real data is abundant. This

hypothesis is tested by whether any of the models fine-tuned with mixed real/synthetic data can outperform the model fine-tuned with all real data.

Experiments on iNaturalist show diverging results between **CoDA** and other baseline augmentation methods: While including synthetic images generated by baseline methods generally led to lower performance, using **CoDA** augmented images can actually lead to improvements over using all real data. Furthermore, a 50-50 real-synthetic data mix generally outperforms all real or all synthetic data. We attribute the success of mixing synthetic and real data to the fact that **CoDA** generated synthetic data is aimed to highlight

| Augmentation Method | Feature Type | LLaVA-NeXT | | | | GPT4o-mini | | | | ViT | | | |
|---|---|---|---|---|---|---|---|---|---|---|---|---|---|
| | | 5:0 | 5:1 | 5:3 | 5:5 | 5:0 | 5:1 | 5:3 | 5:5 | 5:0 | 5:1 | 5:3 | 5:5 |
| Baselines | All Real | **61.2** | - | - | - | **84.3** | - | - | - | **75.4** | - | - | - |
| | Cropping | - | 60.4 | 60.4 | 59.5 | - | 84.8 | 86.3 | 85.9 | - | 78.3 | 77.6 | 79.6 |
| | Flipping | - | 60.7 | 62.9 | 60.1 | - | 83.2 | 83.5 | 84.3 | - | 76.9 | 77.9 | 78.2 |
| | ARMADA | - | 60.7 | 60.2 | 61.2 | - | 84.1 | 84.3 | 83.9 | - | 76.3 | 76.4 | 78.6 |
| CoDA (w/o contrastive) | Textual | - | 74.8 | 75.1 | 74.7 | - | 87.6 | 87.2 | 87.0 | - | 82.5 | 84.5 | 84.7 |
| | Visual | - | 76.5 | 77.9 | 76.2 | - | 88.3 | 89.6 | 88.2 | - | 82.5 | 83.0 | 82.6 |
| | T+V | - | 77.6 | **78.9** | 78.8 | - | 89.5 | **91.2** | 87.9 | - | 84.3 | 84.9 | 82.5 |
| CoDA | Textual | - | 76.4 | 75.9 | 76.8 | - | 87.1 | 87.9 | 87.4 | - | 84.6 | 85.0 | 84.5 |
| | Visual | - | 77.5 | 78.1 | 77.9 | - | 91.3 | 90.8 | **92.6** | - | **85.5** | 84.6 | 85.7 |
| | T+V | - | **78.8** | 78.7 | **79.2** | - | **91.6** | 90.8 | 91.4 | - | 85.3 | **85.8** | **86.3** |

*Table 3.* **Experiments on NovelSpecies with open-weight VLM (LLaVA-NeXT), proprietary LMM (GPT4o-mini), and traditional classifier (ViT) under the Fixed Real Data setting.** Results are in terms of accuracy (%). Synthetic image data generated by Recraft V3. Best performance scores for each setting and scores using all real data are highlighted in **Red** and **Green**, respectively.

discriminable features of the confusing / novel concepts, making them more prominent and visible compared to real images. On the other hand, real images provide valuable style information and is a more accurate reflection of the test-time distribution, helping to "ground" the updated model.

### 5.3. Additional Experiments

For additional experiments, we focus on **NovelSpecies** as it most closely resembles real-world scenarios, where over time, models are required to learn novel concepts without access to sufficient real training data.

**Advanced T2I Model**   As explained in Sec.3, off-the-shelf model components used in **CoDA** can be easily swapped for superior versions of similar models to improve performance. To demonstrate this, we replace the open-weight Stable Diffusion 3.5 Large Turbo model (Stability AI, 2024) with the SOTA proprietary Recraft V3 Model (Recraft.AI, 2024) and run the same LLaVA-updating experiments as in Table 2. Here we note that Recraft V3 has better instruction-following ability as well as better image generation quality compared to Stable Diffusion 3.5 Large Turbo. More details on these differences can be found in Sec.6. Our experiment results in Table 3 show a significant performance boost when LoRA fine-tuning LLaVA with Recraft V3 produced synthetic images compared to fine-tuning on all-real data (28.7%) and also compared to fine-tuning on Stable Diffusion 3.5 Large Turbo produced synthetic data (7.9%). This demonstrates the potential increase of **CoDA**'s effectiveness along with improvements in Text-to-Image generative models. We believe it is also possible to achieve similar improvements by replacing the LLM/VLM components of **CoDA** with superior models in the future.

**Proprietary LMM**   While proprietary LMMs like GPT4o-mini (Hurst et al., 2024) tend to have relatively strong 0-shot performance on existing datasets such as SUN and iNaturalist, their performance significantly degrades on **NovelSpecies** due to having never encountered the novel concepts. To test whether **CoDA** can effectively improve novel concept recognition performance for such proprietary LMMs, we fine-tune the gpt-4o-mini-2024-07-18 model using **CoDA** and relevant augmentation baselines. Results in Table 3 demonstrate a significant performance gain (9.5%) for GPT4o-mini after being fine-tuned on **CoDA** augmented synthetic images. While this improvement is not as significant compared to the LLaVA-1.6 model (20.3%), it is due to GPT4o-mini's better base performance.

**Traditional Classifier**   In addition to evaluating **CoDA** on LMMs which take image-text input and produce text output, we also test whether it can help traditional image classifiers recognize novel concepts. We run the widely-used ViT classifier (Alexey, 2020) on **NovelSpecies** with **CoDA** and other augmentation baselines. Results in Table 3 show that **CoDA** is able to achieve a consistent performance gain over existing baselines for ViT-base (9.1% for single-shot augmentation). The ViT classifier provides stronger base performance compared to general VLMs, thus offering less room for improvement. However, we note here that our main focus on improving LMMs instead of such traditional classifiers stems from LMMs' superior extensibility and generalizability to other related tasks such as recognition-based reasoning and explanation.

## 6. Discussions

In Figure 3, we compare example synthetic images generated by **CoDA** and baseline visual data augmentation methods including Cropping and ARMADA (Jin et al., 2024). While providing localized feature emphasis, Cropping often results in the loss of crucial visual details necessary for concept identification. For instance, for Phyllobates Samperi, cropping occludes the black spots on the frog's skin,

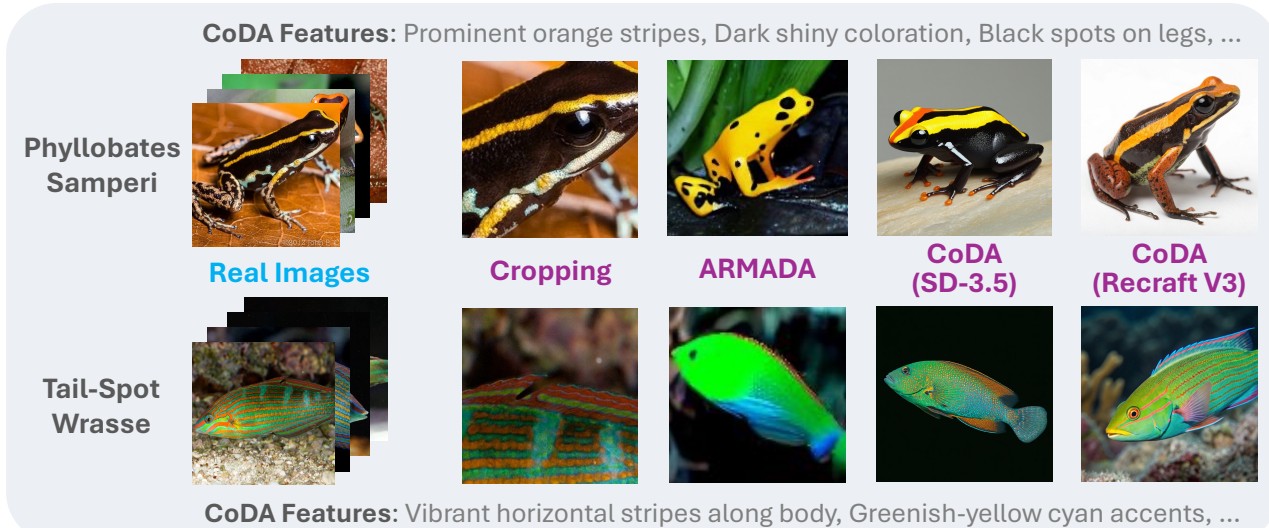

*Figure 3.* **Qualitative Comparison** of **CoDA** and baseline visual data augmentation methods. **Phyllobates Samperi** and **Tail-Spot Wrasse** are example concepts from the **NovelSpecies** dataset. All **CoDA** images are generated using contrastive textual + visual features.

an essential distinguishing feature. Without such essential distinguishing features, the cropped images provide less helpful learning signals compared to other methods.

Unlike Cropping, ARMADA successfully retains some structural features of the target concept, using WikiData text features to guide its image-editing backbone (Brooks et al., 2023). However, this setup also induces two significant issues: **(1)** Leveraging only existing textual features present in WikiData leads to an incomplete feature set, especially for novel concepts. This is apparent in images generated for Phyllobates Samperi, where generated images contained "black spots" but failed to specify their location on the legs of the frog instead of the body. In addition, the model completely failed to generate Phyllobates Samperi's iconic "orange stripes" due to the feature not being recorded in WikiData. **(2)** Image-editing models are not as strong in depicting precise details compared to text-to-image generative models. In the case of the Tail-Spot Wrasse, the ARMADA generated image fails to accurately depict "vibrant horizontal stripes along the body," leading to a visually inconsistent and less biologically accurate representation.

In contrast to existing baselines, Figure 3 shows that **CoDA** is much better at generating high quality synthetic images of the target novel concept that depict accurate and realistic details. Both versions of **CoDA** using different backbone models (Esser et al., 2024; Recraft.AI, 2024) are able to produce significantly more realistic images compared to the two baselines, maintaining general biological consistency. However, we should note that **CoDA**'s performance is inherently bounded by the instruction-following ability of its image generation backbone model, more specifically the

ability to accurately generate multiple feature details in a single image. For example, while CoDA-Recraft-V3 is able to accurately generate all three extracted features including "prominent orange stripes", "dark shiny coloration", and "black spots on legs"; CoDA-SD-3.5 is only able to generate the first two features while failing to capture "black spots on legs". With such limitations in mind, we give **CoDA** an extremely modularized design. This allows each pre-trained model component in **CoDA** to be easily replaced for newer and stronger versions of similar models, including more perceptive VLMs and T2V generative models with stronger instruction-following ability and higher generation quality.

## 7. Conclusion

In this work, we propose **CoDA**, a contrastive visual data augmentation approach that helps LMMs recognize novel, confusing, and low-resource concepts through efficient and effective model updating. **CoDA** is a plug-and-play method which utilizes off-the-shelf models for contrastive feature extraction, feature filtering, text-to-image generation, and image filtering. We evaluate **CoDA** against four existing baselines and self-ablations on three datasets: INaturalist, SUN, and **NovelSpecies**, which we created in this work. Consisting only of animal species discovered in recent years, **NovelSpecies** offers an ideal testbed for LMMs' novel concept recognition. We provide comprehensive additional experiments demonstrating **CoDA**'s effectiveness for traditional classifiers and proprietary LMMs. Finally, we show that **CoDA** can be easily improved by replacing off-the-shelf components, such as text-to-image generation model with superior versions of similar models in the future.

# Acknowledgment

This material is based on research supported by the ECOLE program under Cooperative Agreement HR00112390060 with the US Defense Advanced Research Projects Agency (DARPA), an award from Office of Naval Research with #N00014-23-1-2780, Apple Research Award, and Amazon AGI Research Award. The views and conclusions contained herein are those of the authors and should not be interpreted as necessarily representing DARPA, or the U.S. Government.

# Impact Statement

This paper presents work whose goal is to advance the field of Machine Learning. There are many potential societal consequences of our work, none which we feel must be specifically highlighted here.

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

# A. Appendix

## A.1. Limitations and Future Work

Our work is not without limitations. First, in our experiments, we focus on the fine-tuning use case as it is the most general and intuitive way to utilize our augmented visual data. In the future, we plan to investigate other conceivable use cases for our augmented data, include model adaptation (Sung et al., 2022), test-time augmentation (Gidaris & Komodakis, 2018), visual information retrieval (Wu et al., 2025), and more. Third, the modularity of our method also invites other researchers to replace components of CoDA with superior models to achieve better performance. The NovelSpecies dataset, which can be updated with new species in future years, may also be used to evaluate future VLMs' novel concept recognition abilities. Finally, we also expect improved versions of T2I generation-based visual data augmentation techniques to eventually surpass CoDA in effectiveness and efficiency. Potential improvements may include more robust image / feature filtering and more controllable text-conditioned image generation like multi-view synthesis. We hope our work can pave the way for future downstream advancements by demonstrating effective uses of our augmented visual data for enhancing model capabilities.

## A.2. Data Selection Strategy

For each dataset, we focus on a randomly selected subset of concepts that the model is unable to recognize. The data selection strategy is as follows: In each iteration, we select a random subset of 15 species across different supercategories, including "Birds," "Mammals," and "Reptiles." This strategy allows us to identify confusing pairs without overloading the system, progressively building a collection of challenging cases from each subset. For each species within a subset, we create prompts in a multiple-choice format, incorporating the image and a randomized list of options from all species in the subset. Based on the response from the LMM, we are able to highlight specific species that are commonly mistaken for each other, guiding us in selecting pairs for further analysis. In particular, misclassification happens when an image of one species is identified by the LLM to be an image of another species. A pair $(A, B)$ is considered as a confusing pair if rate of misclassification on either direction is above the threshold 0.2. The process is repeated across new subsets, incrementally building an ample dataset of concepts the model has difficulty recognizing.

## A.3. Experiment Details

### A.3.1. FEATURE EXTRACTION

For textual feature extraction, we use GPT-4o-mini with chain-of-thought reasoning, running with OpenAI API calls. Each API call processes up to 2048 tokens, costing approximately 0.0025 per 1K input tokens and 0.005 per 1K output tokens. Given an average of 500 tokens per query and 10 queries per concept, the estimated cost per concept is around \$0.0375.

For visual feature extraction, we utilize GPT-4o-mini running with OpenAI API calls. Images are preprocessed to a resolution of 336x336 pixels and normalized before feature embedding extraction. Each image query incurs a cost similar to textual feature extraction. With an estimated 5 images processed per concept, the cost per concept amounts to approximately 0.1875.

With the rapid advancement of open-weights large language models and vision language models including DeepSeekV3 (Liu et al., 2024a), DeepSeekVL2 (Wu et al., 2024), Llama 3.2 (Dubey et al., 2024), and more; we expect that feature extraction LLMs and VLMs can be replaced with these models with none or minimal impact to performance. We plan to perform experiments on some of these models and provide comparison results in the next updated version of our work.

### A.3.2. FEATURE FILTERING

We employ CLIP for automatic feature filtering, evaluating Discriminability and Generability scores. Discriminability is computed using cosine similarity between feature embeddings of target and misidentified concepts, with a threshold of 0.6. Generability is assessed by comparing feature presence in synthetic images using an ensemble of Stable Diffusion 3.5 Large and RecraftV3 models. The feature selection step is executed on an NVIDIA A100 GPU, processing features in approximately 2 hours. Top 5 ranked features are selected per concept.

### A.3.3. IMAGE GENERATION AND VERIFICATION

For synthetic image generation, we employ Stable Diffusion 3.5 Large, running on a single A100 GPU. Additionally, we also integrate the RecraftV3 model through an API call. Image generation is performed at a resolution of 512x512 pixels

with a guidance scale of 7.5. The pipeline generates 50 images per concept in approximately 1.2 seconds per image.

Post-generation, we perform automated verification using LLaVA V1.6-34b, running on an A6000 GPU. Each image would take approximately 1 minutes to run for feature presence using a feature-matching confidence threshold of 0.85. Images with a satisfaction rate $S(i^{\text{synthetic}}, \mathcal{F}, \mathcal{M}) < 1.0$ are discarded.

### A.3.4. MODEL UPDATING

We train V1.6-34b with supervised fine-tuning (SFT) using LoRA with rank 128 and alpha 256, optimizing memory efficiency while maintaining model expressiveness. The training runs on two NVIDIA A6000 GPUs, leveraging DeepSpeed Zero-3 for distributed optimization and mixed precision (bf16) for efficiency. The vision encoder is CLIP-ViT-Large-Patch14-336, with an MLP projector aligning visual and text features. We use a cosine learning rate scheduler with a 3% warmup ratio, training for 30 epochs with a batch size of 5 and a learning rate of 2e-4. Images are padded for aspect ratio consistency, and gradient checkpointing is enabled to reduce memory usage. Checkpoints are saved every 50,000 steps, retaining only the most recent one.

### A.3.5. EVALUATION

Automatic evaluation measures zero-shot classification accuracy on a held-out test set. Inference runs on a single A6000 GPU with a batch size of 20, taking approximately 1 hour to complete. The prompt templates for evaluation are attached to Appendix A.4

## A.4. Prompt Construction

```
##  Prompt for Visual/Text Feature Extractions:
# Contrastive Visual
GPT_Contrastive_Visual_Prompt = "You are an experienced and meticulously observant
    biological scientist who is asked to carefully assess the provided image. As labelled
    in the image, the left half of the image contains a picture of the animal {main_class}
     and the right half contains a picture of the animal {confusing_class}. Now, your task
     is summarize the key distinctive visual attributes possessed by {main_class} (on the
    left of the image) that makes uniquely discernible from the {confusing_class} (on the
    right half of the image). Reason step by step to produce an answer. Finally, output
    the key visual attributes of a {main_class} (that make it distinct from a {
    confusing_class}) in a python list format containing short phrases of less than 8
    words each. Do not output any features of the {confusing_class} in your python list.
    Make sure not to name the {main_class} or the {confusing_class} in any of the
    attributes in your list. Also, please try not to use negation in the visual attributes
     you generate: for example, change features like 'lack of facial markings' to 'plain
    brown face'. Additionally, do not use comparative form in any of the features you
    output, for example, change features like 'thinner body than the other class' to 'thin
     body'."

# Visual
GPT_Visual_Prompt = "You are an experienced and meticulously observant biological
    scientist who is asked to carefully assess the provided image. The image contains a
    picture of the animal {main_class}. Now, your task is summarize the key distinctive
    visual attributes possessed by {main_class}. Reason step by step to produce an answer.
     Finally, output the key visual attributes of a {main_class} in a python list format
    containing short phrases of less than 8 words each. Make sure not to name the {
    main_class} in any of the attributes in your list. Also, please try not to use
    negation in the visual attributes you generate: for example, change features like '
    lack of facial markings' to 'plain brown face'. Additionally, do not use comparative
    form in any of the features you output, for example, change features like 'thinner
    body than the other class' to 'thin body'."

# Contrastive Text
GPT_Contrastive_Text_Prompt = "You are an experienced and knowledgeable scene
    classification specialist who is tasked to summarize the key distinctive visual
    attributes possessed by {main_class} that makes uniquely discernible from the {
    confusing_class} (just based on a visual image). First retrieve your knowledge about
    the two different types of scenes, then reason step by step to produce an answer.
```

```
        Finally, output the key visual attributes of a {main_class} (distinct from a {
        confusing_class}) in a python list format containing short phrases of less than 8
        words each. Do not output any features of the {confusing_class} in your python list.
        Make sure not to name the {main_class} or the {confusing_class} in any of the
        attributes in your list. Also, please try not to use negation in the visual attributes
         you generate: for example, instead of saying 'no bright lights,' use 'dark
        environment.' Additionally, do not use comparative forms in any of the features you
        provide. For instance, instead of saying 'smaller windows than the other place,' use '
        small windows.'"

# Text
GPT_Text_Prompt = "You are an experienced and knowledgeable scene classification
    specialist who is tasked to summarize the key distinctive visual attributes possessed
    by {main_class}. First retrieve your knowledge about the {main_class}, then reason
    step by step to produce an answer. Finally, output the key visual attributes of a {
    main_class} in a python list format containing short strings of less than 8 words each
    . Make sure not to name the {main_class} in any of the attributes in your list. Do not
     output any features of the {confusing_class} in your python list. Also, please try
    not to use negation in the visual attributes you generate: for example, instead of
    saying 'no bright lights,' use 'dark environment.' Additionally, do not use
    comparative forms in any of the features you provide. For instance, instead of saying
    'smaller windows than the other place,' use 'small windows.'"

##  Text to Image Generation Prompt
Stable_Diffusion_Text_to_Image_Generation_Prompt = "Generate a 4K realistic image of {
    main_class} that contains the following attributes: " + ', '.join(attributes)

##  Feature Verification Prompt
llava_Verification_Prompt = "You are an image verification specialist. Your task is to
    meticulously assess the image for specific attributes and confirm their presence. For
    each attribute in the list, carefully check the image, examine visual elements such as
     color, shape, texture, position, and context clues that might indicate whether the
    attribute is present. Provide a binary python output list, where each element is
    either 1 (attribute is present) or 0 (attribute is absent), corresponding exactly to
    the order of attributes provided.\\n\\nAttributes to Verify:{attributes}\\n\\nExpected
     Output: A list of 0s and 1s indicating the presence or absence of each attribute, in
    the same order as listed. Here is an example output: [0, 1, 1]."

##  Finetune and Evaluation Prompt
llava_Finetune_and_Evaluation_Prompt = "You are an image classification specialist with
    expertise in categorizing images into specific groups. Given an image, identify its
    category from the following options: " + ", ".join(provided_options_capitalized[:-1])
    + ", or " + provided_options_capitalized[-1] + ". Provide your answer as only one
    category name for precise classification. Please response with the category name only.
    "

##  Deduplication Prompt
GPT_Deduplication_Prompt = "You are an experienced and knowledgeable biological scientist
    who is tasked to summarize the key distinctive visual attributes possessed by {
    main_class} into a coherent list. Given the following list of attributes describing
    the animal species {main_class}: {attributes_list}. You task is to combine the
    duplicate features (which have the same or very similar meanings) into one. Then, you
    will order the remaining features in order of visual importance, the most visually
    significant / observable features will be at the front of the list while the least
    visually observable features will be at the back. Finally, output the key visual
    attributes of a {main_class} in a python list format containing short phrases of less
    than 8 words each. Make sure not to name the {main_class} in any of the attributes in
    your list. Also, please try not to use negation in the visual attributes you generate:
     for example, change features like 'lack of facial markings' to 'plain brown face'.
    Additionally, do not use comparative form in any of the features you output, for
    example, change features like 'thinner body than the other class' to 'thin body'."

##  System Prompt
GPT_System_Prompt = "You are a helpful assistant."
```

