# OpenReview forum: "Contrastive Visual Data Augmentation"
_ICML.cc/2025/Conference — ICML 2025 poster_

### Official Review · Reviewer_DV9x · 2025-03-10

**Overall Recommendation:** 3

**Summary:**

This paper proposes a novel data augmentation technique aimed at improving the recognition capabilities of Large Multimodal Models (LMMs) on rare classes/concepts that are underrepresented in the training set. In particular, authors leverage text-to-image diffusion models to synthesize images of the rare concept, where the generation prompt is crafted via a contrastive technique focused on highlighting the differences between the rare concept and its similar, but common counterpart. Experiments demonstrate that if only a few (5-10) examples of the rare concept are available for fine-tuning, the proposed technique can significantly boost recognition performance over some simple baselines.

## update after rebuttal
I thank the authors for clarifying my questions and concerns. I raised my score accordingly.

**Claims And Evidence:**

- The claim in the contributions list "CoDA is also the first widely successful method using text-to-image generation for visual data augmentation" in my opinion is too strong. T2I diffusion models have been used for data augmentation before (see ARMADA [1], a baseline from the paper, TTIDA [2] and a survey paper [3]) and "widely successful" is subjective.

- Authors claim that CoDA "significantly improve data and compute efficiency compared to existing methods", and while I can get behind the data efficiency aspect (with the same number of real samples, CoDA achieves higher accuracy), compute efficiency is not demonstrated in any way in the paper. In fact, it appears that the technique requires significant effort, involving several steps of feature extraction using LLMs and frontier diffusion model-based generation. Comparative study on the cost of the technique is missing.

[1] Jin, Xiaomeng, et al. "ARMADA: Attribute-Based Multimodal Data Augmentation." Proceedings of the First Workshop on Advancing Natural Language Processing for Wikipedia. 2024.

[2] Yin, Yuwei, et al. "Ttida: Controllable generative data augmentation via text-to-text and text-to-image models." arXiv preprint arXiv:2304.08821 (2023).

[3] Alimisis, Panagiotis, et al. "Advances in diffusion models for image data augmentation: A review of methods, models, evaluation metrics and future research directions." Artificial Intelligence Review 58.4 (2025): 1-55.

**Essential References Not Discussed:**

I believe the references are sufficient to understand the context and contributions.

**Experimental Designs Or Analyses:**

I have some concerns with respect to the experimental evaluation.

First, the primary dataset, NovelSpecies, is very small. The results are reported over 64 datapoints, which casts some doubt on the statistical significance of the reported results.

Second, the improvements are very inconsistent: in some experiments textual features alone are the best, in other cases visual, and in some cases the combination of both. In some cases the proposed contrastive feature extraction helps, in other cases it doesn't. I believe there has to be more thorough study on the role of the different components of the pipeline, and some guidelines how to select which arrangement to use. The large variance in results may also be a result of the small size of the dataset.

Third, it would be necessary to add some naive baselines to gauge the effectiveness of the technique. What happens if we use few-shot prompting with or without augmented samples? What happens if we use few-shot prompting highlighting the differences in text format?

Lastly, experiments on larger scale datasets are only performed using a single model (LLaVA-1.6), casting some doubt on the generality of the results.

**Methods And Evaluation Criteria:**

The proposed approach is sensible. Explicitly highlighting the differences between known concepts and a novel concept can be an efficient way to learn the characteristic attributes of the novel concept in a data-efficient way.

One caveat I have with the methodology is that it is bottlenecked by the VLMs/LLMs used for extracting visual/textual features. If the concept is truly novel, it is possible that the defining attributes cannot be extracted this way (as the feature extractors haven't seen the concept during training either). In other words, the proposed technique only works in cases where the novel concept is defined by a unique/novel combination of already known attributes. However, for recognizing truly novel concepts, external knowledge may be required.

Another concern I have is the current limitations of text-to-image models in correctly generating multiple attributes based on the input prompt, which is crucial for the technique. In particular, [1] shows that even SOTA proprietary T2I models fail to generate as low as 3 distinct attributes correctly in 50% of the cases, and the success probability approaches 0 for 7 attributes. Thus, I believe the granularity of how much contrastive information can be represented in the synthetic images is seriously limited by the current performance of T2I models.

Finally, I am unsure about using CLIP for calculating Discriminability/Generability scores. In particular, CLIP may have the exact same bias as the LMM towards the well-represented concept and thus  may more frequently associate attributes with known concepts simply due to training set bias.

[1] Wu, Xindi, et al. "Conceptmix: A compositional image generation benchmark with controllable difficulty." arXiv preprint arXiv:2408.14339 (2024).

**Other Comments Or Suggestions:**

I found the term "hyper-domain-specific" verbose and vague at the same time. How is it defined and how is it different from domain-specific?

**Other Strengths And Weaknesses:**

I believe that the core idea of the paper is original and sensible. The significance of the findings is somewhat diminished due to (1) limited dataset size, limited number of models, (2) lack of discussion on cost/compute and (3) lack of ablation studies on the different components of the method. The modularity of the approach and the potential to improve with the emergence of better models is a nice benefit. Moreover, adapting LMMs to novel concepts efficiently is an important direction giving strong motivation to the paper. I find the writing to be more or less clear.

**Questions For Authors:**

See my points under Other Strengths and Weaknesses.

**Relation To Broader Scientific Literature:**

The paper is closely related to and advances the work proposed in [1] by performing the editing in a more targeted way: the edit highlights the features that are different in the novel class from a common class the model confuses it with.

[1] Jin, Xiaomeng, et al. "ARMADA: Attribute-Based Multimodal Data Augmentation." Proceedings of the First Workshop on Advancing Natural Language Processing for Wikipedia. 2024.

**Theoretical Claims:**

No proofs/theoretical claims.

---

> ### Author Rebuttal · Authors · 2025-04-01
>
> We are pleased that reviewer DV9x finds:
> * our **idea original and sensible**
> * our **technique novel**
> * our **references sufficient**
> * and the **modularity of our approach beneficial**
>
> We value your constructive comments and address them in detail:
>
> ---
> **Related Data Augmentation Methods**
>
> We want to highlight a key distinction between our **generation-based** and existing **editing-based** augmentation (ARMADA and all methods in [1]): While our method generates T2I augmented images from scratch, editing-based methods create augmented images by perturbing attributes of real images of other concepts (L94-105), leading to two drawbacks:
> - they require large amounts real images, which are rare for novel concepts.
> - augmented images often lack background and view variation.
>
> Experiments in Table 2 show that fine-tuning with such data yield very limited gains, on par with traditional editing-based augmentation methods like Flipping.
>
> Furthermore, while we aim to thoroughly compare with **published** related works such as ARMADA, at the time of submission, both TTIDA and [1] were **not published** in any known conference or journal. Nevertheless, we are happy to cite and thoroughly discuss both works in our camera ready version.
>
> ---
> **Can VLMs Describe Features of Truly Novel Concepts**
>
> While editing-based methods can only create augmented images by perturbing known attributes present in existing concepts, our generation-based approach can design any novel feature to describe novel concepts. Therefore, **this limitation only exists for editing-based methods**, while CoDA is able to create any novel feature that can be described with human language.
>
> Furthermore, Chiquier et al. [2] showed modern VLMs can generate high-quality novel descriptive features for rare species and even otherworldly concepts such as those in the KikiBouba Dataset.
>
> ---
> **Clarification on Compute Efficiency**
>
> We'd like to clarify: By "compute efficiency", we refer to **efficiency during fine-tuning** with augmented / real images. As described in L631-654, CoDA is inference-only and does not involve any expensive back-propagation. Thus, CoDA's compute cost is an order of magnitude less than the fine-tuning process.
>
> Our efficiency claim refers to: when comparing models fine-tuned with CoDA augmented images vs models fine-tuned with all real images or ARMADA images, models fine-tuned with CoDA images can achieve higher performance with lower fine-tuning cost. Table 2 results justify this: training on 6 images per concept of mixed Real+CoDA data outperforms training with 20 real images per concept (using over 2x the compute).
>
> We acknowledge the word 'compute efficiency' was ambiguous in the paper and will clarify it in the camera-ready version.
>
> ---
> **Can T2I Models Generate Multiple Attributes**
>
> This is an issue we encountered many times. Consistent with Conceptmix, we found older T2I models like SD-2.1 tend to fail at generating >3 attributes in a single image. However, newer T2I models not included in the benchmark, like SD-3.5, can reliably generate >5 concepts.
>
> Furthermore, our two-layer feature filtering (L171-209) and image filtering (L236-251) strategy is specifically designed to filter out any remaining generation failures. Our human evaluation in (Table 1) demonstrates that the vast majority of features (83.97%) were successfully generated:
>
> |Image Type|Feature Presense(%)|IAA(κ)
> |-|-|-
> |Real|92.51|0.87
> |Synthetic|83.97|0.82
>
> ---
> **CLIP Discriminability/Generability scores**
>
> Our intentional math formulation of Discriminability/Generability scores ensure that the proposed association bias of CLIP would actually **aid in filtering out non-discriminative features**, while having no impact on the relative ranking of generability scores. Due to word limits, please respond for a detailed breakdown.
>
> ---
> **Dataset Size and Generality of Results**
>
> We want to clarify: while the NovelSpecies dataset has **64 different species**, it actually contains **2240 annotated images**. Results on NovelSpecies is reported on val and test sets each containing **960 unique images** (L275-277). We tested **three models** on this dataset to show CoDA's generality (Table 3).
>
> To further prove the statistical significance of our NovelSpecies results, we ran 5 independent sets of experiments on 10 different settings of NovelSpecies with different random seeds:
>
> Results show remarkable consistency: **average standard deviation of scores is 0.0133** across 10 settings, with a max of 0.017. The **average coefficient of variation is 1.82%** across all settings, **far below the general 5% threshold** for statistical stability.
>
> |Statistic|Min|Max|Avg
> |-|-|-|-
> |Standard Deviation|0.0069|0.0170|0.0133
> |Coefficient of Variation|0.0087|0.0249|0.0182
>
> (Respond to see  full score table)
>
> [1] Advances in diffusion models for image data augmentation. AI Review (2025)
>
> [2] Evolving Interpretable Visual Classifiers with Large Language Models. ECCV (2024)

---

> > ### Comment · Reviewer_DV9x · 2025-04-07
> >
> > I thank the authors for clarifying some of my doubts and concerns.
> >
> > However, I am confused why CoDA would be "inference-only and does not involve any expensive backpropagation". In all experiments, the base model is fine-tuned on the synthetic data. I maintain that feature extraction and image generation using foundation models as well as SFT incurs non-trivial compute cost and it has to be carefully compared to the cost of other comparable techniques in order to substantiate any claims on compute-efficiency.
> >
> > Furthermore, my comments on naive baselines is not addressed. What if we use few-shot prompting techniques without fine-tuning? I am especially interested in how models perform by (1) simply adding a single contrasting example in the prompt to show the model the visual differences and (2) simply adding textual description of the visual differences in the prompt. These are computationally cheaper and would be interesting to see how much CoDA can improve over these sensible baselines.

---

> > > ### Author Response · Authors · 2025-04-09
> > >
> > > We sincerely thank Reviewer DV9x for their engagement and thoughtful feedback. Your comments have been helpful in refining our work. Below, we address your points regarding compute efficiency and naive baselines.
> > >
> > > ---
> > >
> > > ### **"CoDA is inference-only and does not involve any expensive back-propagation"**
> > >
> > > Here we would like to clarify a potential misunderstanding: **CoDA itself specifically refers to our synthetic data generation process**, which only includes the following components:
> > >
> > > - Feature extraction with VLM inference
> > > - Image generation with T2I model inference
> > > - Feature and image filtering with CLIP and VLM inference
> > >
> > > The fine-tuning process in our experiments is a downstream usage of CoDA generated synthetic data, not **part of the CoDA method** itself.
> > >
> > > ---
> > > ### **Further Clarifications regarding Efficiency**
> > >
> > > We **`do not claim`** that CoDA's synthetic data generation process itself is more efficient compared to other synthetic data generation processes (which is not true as crop/flip requires no compute at all).
> > >
> > > Instead, we claim that: fine-tuning with CoDA augmented images yields higher performance with lower fine-tuning cost compared to fine-tuning with images generated by other data augmentation techniques. This is justified by results from Table 2: training on 6 images per concept of mixed Real+CoDA data outperforms training with 20 real images per concept (which costs over 2x the compute).
> > >
> > > The reviewer mentions a third method of comparison: i.e. comparing the data generation cost + fine-tuning cost of CoDA vs that of other methods. In the paper, we also **`do not claim`** that this comparison would show CoDA to be more efficient.
> > >
> > > We recognize that the distinction between these three methods of comparison was not made abundantly clear in our original submission. Therefore, we will remove all claims regarding compute efficiency in our camera-ready version to avoid any misunderstandings.
> > >
> > > ---
> > > ### **Naive Baselines**
> > >
> > > The reviewer suggested two naive inference-time augmentation baselines:
> > > 1. adding a single pair of contrasting image example in the prompt
> > > 2. adding textual description of the concept differences in the prompt
> > >
> > > During the exploration phase of our work, we considered these baselines but identified **three major issues severely limiting their general applicability**:
> > >
> > >
> > > - Both methods require the target model to be able to accept interleaved image-text input. Traditional classifiers like ViT, which excel in single-image tasks (Table 3), cannot leverage these methods.
> > > - Both methods are only applicable for binary classification. In general multi-way classification with more than two choices, it is impossible to pre-determine which set of example images or textual description should be provided to the model.
> > > - Method 1 can only work for VLMs that have strong multi-image referring and reasoning abilities. However, most VLMs at the time were not capable enough, as shown by a binary classification experiment with LLaVA 1.6 32B, yielding near-random performance when given image examples:
> > >
> > > |Dataset|1-shot Acc(%)|3-shot Acc(%)|Random Acc(%)
> > > |-|-|-|-
> > > |NovelSpecies|48.6|50.7|50.0
> > > |INaturalist|49.3|47.9|50.0
> > >
> > > However, given the reviewer's suggestion and recent advancements in VLMs' multi-image reasoning ability, we ran the following binary classification experiments on the gpt-4-turbo-2024-04-09 model:
> > >
> > > |Dataset|0-shot Acc(%)|CoDA Image 1-shot Acc(%)|CoDA Text Features Acc(%)|CoDA Text Features + CoDA Image 1-shot Acc(%)
> > > |-|-|-|-|-
> > > |NovelSpecies|87.88|91.58|91.32|95.26
> > > |iNaturalist|84.08|87.63|86.71|88.55
> > >
> > > Results demonstrate notable improvements when using 1-shot CoDA augmented example images and/or CoDA text features during inference. This shows that **CoDA can indeed help improve model performance via inference-time augmentation techniques**, although this improvement is only observed for binary classification tasks with strong VLMs (due to limited time and significant compute costs, we did not try more settings). We will add this result to our camera-ready paper as it is an interesting finding although not within the main focus / claimed contributions of our work.

---

### Official Review · Reviewer_bmru · 2025-03-13

**Overall Recommendation:** 4

**Summary:**

In this paper, authors proposes Contrastive Visual Data Augmentation (CoDA), a novel approach to improve LMMs' ability to recognize novel and easily confused visual concepts. CoDA extracts contrastive features between target concepts and the confusable counterparts, generating synthetic training data with test-to-image models to update LMM. This paper also contributes a novel dataset, NovelSpecies. The solid experimental results demonstrate the effectiveness of CoDA.

**Claims And Evidence:**

Yes, they are.

**Essential References Not Discussed:**

Authors have cited and discussed related works.

**Experimental Designs Or Analyses:**

Yes, I did.

**Methods And Evaluation Criteria:**

Yes, the proposed method makes sense for the problem.

**Other Comments Or Suggestions:**

There are some typos in the paper.
1. For cross reference in the paper, Fig.1, and Tab.1 should be shown as Fig. 1, and Tab. 1 or Figure 1 and Table 1, where there is a space between "." and the number.
2. More visualizations like Figure 3 can be presented in the Appendix.
3. I'm wondering whether up-to-date online large models (such as GPT-o3 mini) could recognize the novel concepts. (There is no requirement for this point during the rebuttal.)

**Other Strengths And Weaknesses:**

Strengths:
1. The paper is well-written to follow.
2. The proposed data augmentation strategy is effective for novel concepts.
3. The NovelSpecies dataset is interesting.
4. Competitive performances.
---
From my perspective, there are no obvious weaknesses of this paper.

**Questions For Authors:**

Please see above.

**Relation To Broader Scientific Literature:**

1. This paper studies the data augmentation strategy, which is extensively applied for model training.
2. A novel benchmark for current LMMs.

**Theoretical Claims:**

There are not theoretical claims in this paper.

---

> ### Author Rebuttal · Authors · 2025-04-01
>
> We sincerely thank Reviewer bmru for their insightful and encouraging comments. We are pleased that you found:
> * our paper **well-written to follow**,
> * our proposed **data augmentation strategy effective**,
> * our **method's performances competitive**,
> * and our new **NovelSpecies dataset interesting and novel**.
>
> We are also encouraged that you recognize that our **solid experimental results demonstrate the effectiveness of CoDA**.
>
> Please find your suggesion and additional feedback carefully addressed below:
>
> ---
> **Spacing Between Figure and Table Notation**
>
> Thank you very much for catching this formatting issue, your detailed efforts are greatly appreciated. We fully agree that correcting this notation throughout our work would improve the readibility and clarity of our work. Therefore, we will make sure to correct all such usage instances in our camera ready version.
>
> ---
> **More Visualizations like Figure 3 in Paper**
>
> We fully agree with the reviewer that visualizations like Figure 3 can help illustrate our method and provide straightforward comparisons with other related methods. Moreover, we currently plan to include an additional figure in our camera ready version to provide example of success and (rare) failure cases in our visual data generation process. We believe this will help readers better understand the abilities and potential limitations of our work and facilitate better method adaptation to more domains.
>
> ---
> **Whether Online Large Models can Recognize Novel Concepts**
>
> This is a very interesting question raised by the reviewer. We believe that indeed online models with access to up-to-date internet information may be better suited for recognizing novel concepts compared to static models. However, when qualitatively testing current online VLMs such as GPT4o and Claude 3.7 Sonnet, we find that they still fail to recognize most of the concepts in our NovelSpecies Dataset. We believe this is likely due to their reliance on textual retrieval and inability to retrieve the relevant information based on novel images.

---

> > ### Comment · Reviewer_bmru · 2025-04-06
> >
> > I have read the rebuttal and my concerns are addressed. I keep my rating.

---

### Official Review · Reviewer_pV1R · 2025-03-13

**Overall Recommendation:** 3

**Summary:**

The paper proposes a data augmentation technique for the tuning of large multimodal models on unseen concepts that are in a way 'very close' to known concepts included in the training. The method thus aims at expanding the knowledge of an existing model, including new concepts when they are encountered.
The augmentation strategy works on a contrastive learning basis, including text descriptions and generated images of the novel concepts. From this perspective, the paper has a novel approach to adapting (normally done via fine-tuning) models to new concepts that have not been seen during training. Next to the methodology, the authors present a new dataset of animal species, recently discovered, that can be used to effectively test novel concepts wrt the iNaturalistic dataset.

**Claims And Evidence:**

The authors claim that the proposed NovelSpecies dataset can be used to evaluate the ability to recognize novel concepts, which seems extensively demonstrated in the experimental section.
Furthermore, a methodological claim that the method proposed can adapt existing models in an effficient and effective way, thus overcoming the ineffectiveness of methods based on fine-tuning due to the scarcity of data usable for model adaptation. However, comparisons to finetuning-based methods are not evident in the paper. This prevents from appreciating the efficiency and effectiveness contributions claimed.

**Essential References Not Discussed:**

I do not think there are particular missing references. I think that discussing the relation of this paper with test-time augmentation methods (and cite for instance seminal papers on the topic) would make the placement within the literature more solid.

**Ethical Review Concerns:**

The authors perform an evaluation involving humans evaluators - no statements regarding an approval of the ethics committee of the institution is included in the paper neither in the supplementary material.
Questions arise on what data of the humans were recorded, how the tests were submitted to the annotators, and what questions were asked.

**Ethical Review Flag:**

Flag this paper for an ethics review.

**Ethics Expertise Needed:**

["Responsible Research Practice (e.g., IRB, documentation, research ethics, participant consent)"]

**Experimental Designs Or Analyses:**

The chosen datasets and the proposed one are solid choices to provide an extensive validation.
The choice of the comparison methods could be extended with fine-tuning based approaches, as in the introduction they are claimed to be ineffcient and ineffective, two shortcomings that the proposed method addresses directly.
The ablation with different backbones provides extra evidence that the method can be plugged in different architectures.

**Methods And Evaluation Criteria:**

The method is described in a detailed way. It makes sense to use text-guided generative models to expand the data available for augmentation at test-time. The filtering of features also seems to stabilize the adaptation phase, avoiding that the model learns degenerate representations. Overall the method seems fine and is evaluated on several datasets using reasonable metrics. In my opinion, it is novel and provides concrete contributions to the state of the art.

**Other Comments Or Suggestions:**

When use listings (e.g. 1)..., 2)...) there should be no dot '.' after the parenthesis ')' and the item should not start with a capital letter - that's really confusing while reading.

to be explicit
"straightforwardly 1). Fine-tune text decoder on new textual"
should be
"straightforwardly 1) fine-tune text decoder on new textual"

**Other Strengths And Weaknesses:**

The paper is clear, with potentially impactful contributions in the field.
As a weakness, I would say that the efficiency/effectiveness benefits should be better highlighted, as this is key to provide impact.

**Questions For Authors:**

- How does the proposed approach relate to test-time augmentation methods?
- what is the effectiveness/efficiency gain wrt fine-tuning adaptation?

**Relation To Broader Scientific Literature:**

The paper seems well-placed in the current literature and addresses an important problem related to large multimodal models and their deployment in real scenarios. The paper relates, in my opinion, to the literature on test-time augmentation that is used to adapt models at test time. A coverage of this relation should be present in the paper, or the authors should motivate why not.

**Theoretical Claims:**

I did not find theoretical claims or proofs to validate. The paper makes hypotheses and verifies them with a mostly empirical approach.

---

> ### Author Rebuttal · Authors · 2025-04-01
>
> We sincerely thank Reviewer pV1R for their detailed review. We are delighted you recognize that:
> * our **method is novel and provides concrete contributions to the state of the art**,
> * our **claims are extensively demonstrated**,
> * our **datasets are solid choices to provide extensive validation**
>
> We are also encouraged that you find our paper **well-placed in the current literature, addressing an important problem**.
>
> We have carefully considered your valuable feedback and suggestions for improvement, which we address in detail below:
>
> ---
> **Comparison to Fine-tuning Methods**
>
> We very much appreciate the reviewer bringing up this important point and would like to take this opportunity to clarify: While a main contribution of our work is providing a state-of-the-art visual data augmentation strategy, we leave downstream users the freedom to decide how to use our augmented synthetic visual data, whether it is model adaptation, few-shot learning, fine-tuning, or even pre-training. **In our experiments, we focus extensively on the fine-tuning use case as it is the most general and intuitive way to utilize our augmented visual data and demonstrate its general usefulness for improving different models.**
>
> Specifically, we compare **fine-tuning existing models with all real data, mixed real and synthetic data from baseline data augmentation strategies, and mixed real and synthetic data from our CoDA method**. Results in Table 2 and Table 3 show that basic fine-tuning with all real data or including synthetic data from baseline data augmentation strategies produces much lower performance compared to fine-tuning with mixed real and synthetic data from our CoDA method. (L330-420)
>
> For better reference, we paste the following results subset from Table 2 of our main paper to illustrate the **performance disparity of basic fine-tuning with all real data vs including one synthetic image per concept from baseline data augmentation strategy ARMADA and our CoDA method**:
>
> |Dataset| Basic Fine-tuning Acc(%) | ARMADA Acc(%) | CoDA Acc(%) |
> |---|---|---|---|
> | NovelSpecies  | 61.2 | 60.7| 70.1|
> | SUN           | 73.4 | 75.9| 83.4|
> | INaturalist   | 49.2 | 60.1| 63.5|
>
> ---
> **Relation to Test-time Augmentation Methods**
>
> We thank the reviewer for pointing out the relations between our work and existing test-time augmentation techniques, a topic we extensively considered and explored during our project. As elaborated in our response to Issue #1 above, we position our work as providing a state-of-the-art visual data augmentation strategy, while allowing downstream users to decide whether to use our augmented data for pre-training, fine-tuning, or test-time augmentation.
>
> **To validate the usefulness of our augmented data, we currently focus on the fine-tuning scenario because:**
> - General applicability, as fine-tuning is generally applicable to LMMs as well as traditional classifiers.
> - Empirical effectiveness, as the difficulty of the visual novel/confusing concept recognition task limits the effectiveness of test-time augmentation methods.
>
> In fact, during our exploration stage, we ran multiple test-time few-shot generalization experiments on VLMs. Here we attach our **earlier results for 1-shot and 3-shot generalization of LLaVA 1.6 34b, under the extremely simplified binary classification setting**:
>
> |Dataset|1-shot Acc(%)|3-shot Acc(%)|Random Acc(%)
> |-|-|-|-
> |NovelSpecies|48.6|50.7|50.0
> |INaturalist|49.3|47.9|50.0
>
> The model achieves no better than random chance performance on the three datasets with few-shot learning. Further qualitative experiments on proprietary VLMs also showed little promise, which prompted us to move towards fine-tuning methods for better performance.
>
> However, we recognize that the reviewer's comment regarding test-time augmentation methods goes beyond basic few-shot learning. We have identified several **areas within multimodal test-time augmentation methods that may be promising given our CoDA method or relevant to the novel/confusing concept recognition task:**
>
> - Multimodal meta learning and fast adaptation methods such as Flamingo
> - Code-based multimodal reasoning methods including VisProg and ViperGPT
> - Multimodal reflection and self-critique methods including LLaVA-Critic, Critic-V, and VDebugger
> - RL-based inference time scaling methods such as Visual-RFT and VAGEN
> - Multimodal RAG methods including Wiki-LLaVA, UDKAG and UniRAG
>
> We will include discussions relating our method to these test-time augmentation strategies in the camera ready version.
>
> ---
> **Highlighting the Effectiveness/Efficiency Gains**
>
> We thank the reviewer for recognizing the substantial effectiveness / efficiency gains of CoDA demonstrated in the experimental section. To highlight these gains, **we will include an additional bar chart comparing performances of CoDA against other baseline methods.** We believe this visual comparison will help readers more easily grasp the effectiveness / efficiency gains of CoDA.

---

> > ### Comment · Reviewer_pV1R · 2025-04-05
> >
> > I have read the response to my comments and also those to the comments of other reviewers.
> > I thank the authors for their responses, which mostly clarify my questions and doubts.
> > However, my observations regarding test-time adaptation is vaguely addressed. If the authors have extensively considered it in the preparation process of this research project, why it is not discussed? Also, if the authors claim that the proposed approach is more general than test-time augmentation methods and can be used for pre-training, fine-tuning, TTA, etc. how are they going to address and clarify this in the paper?
> >
> > I find the paper overall containing good contributions and possibly having some impact in the field. However, a few things ar eleft hanging, also in the rebuttal.

---

> > > ### Author Response · Authors · 2025-04-09
> > >
> > > We thank Reviewer pV1R for engaging in constructive discussion and recognizing our work's **good contributions and potential impacts to the field**. Below we address your follow-up questions in detail.
> > >
> > > ---
> > >
> > > ### **Scope Clarifications: CoDA is a Data Augmentation Method that does not include Model Updating or Inference**
> > >
> > > Our proposed technique **CoDA**, is a **`visual data augmentation method`** designed to generate high-quality synthetic data, which may potentially be used as input to downstream model updating and inference methods such as:
> > > - Fine-tuning
> > > - Pre-training
> > > - Test-time Augmentation (TTA)
> > > - Adapters
> > >
> > > CoDA itself is **not** a **`model updating / inference method`**, and we **do not claim** any novel contributions in model updating techniques. Therefore, CoDA should only be compared with other **`visual data augmentation methods`** (e.g. ARMADA) instead of other **`model updating / inference methods`** (e.g. adapters or TTA). We also **do not claim** that our method will be useful for all downstream model updating / inference methods, as it is impossible to exhaustively verify within a single paper.
> > >
> > >
> > > To demonstrate the usefulness of CoDA generated visual data, we run experiments on fine-tuning VLMs and Traditional Classifiers using CoDA generated images. We **do not claim** that CoDA is more general than TTA methods (they are not comparable), but rather that we chose fine-tuning over TTA to showcase CoDA's usefulness because we found fine-tuning to be more generally applicable for our specific task. (discussed below)
> > >
> > > ---
> > > ### **Considerations Regarding Test-time Augmentation**
> > >
> > > During our work's exploration stage, we considered two inference-time augmentation strategies:
> > > 1. adding few-shot image examples in the prompt
> > > 2. adding textual descriptions of the concept differences in the prompt
> > >
> > > However, there were three major issues limiting the general applicability of these TTA methods compared to fine-tuning-based model updating methods:
> > >
> > > - Both methods require the target model to be able to accept interleaved image-text input. Traditional classifiers like ViT, which excel in single-image tasks (Table 3), cannot leverage these methods.
> > > - Both methods are only applicable for binary classification. In general multi-way classification with more than two choices, it is impossible to pre-determine which set of example images or textual descriptions should be provided to the model.
> > > - Method 1 can only work for VLMs that have strong multi-image referring and reasoning abilities. However, many VLMs at the time were not capable enough, as shown by a binary classification experiment with LLaVA 1.6 32B, yielding near-random performance when given image examples:
> > >
> > > |Dataset|1-shot Acc(%)|3-shot Acc(%)|Random Acc(%)
> > > |-|-|-|-
> > > |NovelSpecies|48.6|50.7|50.0
> > > |INaturalist|49.3|47.9|50.0
> > >
> > > Given the reviewer's suggestion and recent advancements in VLMs' multi-image reasoning ability, we additionally tried the following binary classification experiments on the newer gpt-4-turbo-2024-04-09 model:
> > >
> > > |Dataset|0-shot Acc(%)|CoDA Image 1-shot Acc(%)|CoDA Text Features Acc(%)|CoDA Text Features + CoDA Image 1-shot Acc(%)
> > > |-|-|-|-|-
> > > |NovelSpecies|87.88|91.58|91.32|95.26
> > > |iNaturalist|84.08|87.63|86.71|88.55
> > >
> > > Results demonstrate notable improvements when using 1-shot CoDA augmented example images and/or CoDA text features during inference. This shows that CoDA can indeed help improve model performance via inference-time augmentation techniques, although this improvement is only observed for binary classification tasks with strong VLMs. We will add this result to our camera-ready paper as it is an interesting finding to motivate future work, althought not within the main focus or claimed contributions of this paper.
> > >
> > > The above results are only **additional explorations** intended to provide more insights into the reviewer's suggestion regarding consideration of TTA methods. We **do not claim** to have exhaustively studied all TTA methods for our task, and leave the task for future works. We welcome the reviewer to propose any additional methods that can be potentially helpful, and we will be happy to cite/discuss them in our camera-ready version.
> > >
> > >
> > > ---
> > > ### **Summary**
> > > In summary, CoDA is a visual data augmentation method, validated through fine-tuning-based experiments. We did not propose a new model updating / inference technique comparable to TTA, so our evaluation rightly focused on fine-tuning to measure CoDA’s impact. We will make this scope explicit in the paper and will acknowledge TTA as a promising future direction.

---

### Official Review · Reviewer_Xmi8 · 2025-03-19

**Overall Recommendation:** 4

**Summary:**

The current submission addresses a known issue in LMMs (Large Multimodal Models) that of recognizing novel or even confusing visual concepts, due to their reliance on pre-trained knowledge and their limited ability to capture subtle visual details. To this en,d the authors introduce CoDA (Contrastive visual data augmentation) that extracts key contrastive both textual and visual features that differentiate target concepts from concepts they are commonly confused with. Afterward, text-to-image generative models are used to create synthetic training data that highlight these distinctive features, automatically filtering these images for quality assurance. The authors evaluate CoDA on three datasets: iNaturalist, SUN, and a newly introduced dataset called NovelSpecies, consisting of recently discovered animal species guaranteed to be unseen by the LMMs. CoDA is shown to generalize well across proprietary LMMs (GPT4o-mini) and traditional classifiers (ViT) and makes non-trivial improvements in accuracy over state-of-the-art augmentation methods in all the tested scenarios.

**Claims And Evidence:**

Claims are well-supported by evidence, such as:
* CoDA outperforms existing visual data augmentation methods demonstrated through comparative evaluations on three datasets
* CoDA is more effective in dealing with novel concepts, proved by the experiments conducted on a novel, dedicated and challenging benchmark for this task
* broad applicability to different models such as ViTs and proprietary LMMs (GPT4o-mini)
* solid ablation studies that prove the effectiveness of contrastive feature extraction and augmented image filtering (human evaluation is also used)

**Essential References Not Discussed:**

The authors have cited a broad range of relevant literature, and I did not find any glaring omissions of essential works.

**Experimental Designs Or Analyses:**

The conducted experiments are sound and well-executed and include:
1. CoDA comparison against multiple baselines on three datasets
2. CoDA variations (textual-only, visual-only, both)
3. Test on different model architectures
4. Ablation studies to validate key components
5. Human evaluations to verify feature and image quality

**Methods And Evaluation Criteria:**

The evaluation criteria and benchmarks are appropriate. The authors evaluate on well-known benchmarks (iNaturalist, SUN) and a novel dataset (NovelSpecies) specifically designed for evaluating the recognition of novel concepts that are guaranteed to be outside the training data of any LMM (with a knowledge cutoff date).

**Other Comments Or Suggestions:**

* A more detailed analysis of the computational overhead of CoDA compared to other methods would be useful for practitioners considering adoption.
* Overall, I have very few critiques on writing or execution (I did not catch any obvious typos).

**Other Strengths And Weaknesses:**

Overall, the strengths of this paper far outweigh the weaknesses.

Strengths:
* The paper addresses a practical and important problem for LMMs in a novel way
* I fairly enjoyed reading the submission, well-written, well-structured, good discussion
* The creation of NovelSpecies dataset is a valuable contribution to the field for benchmarking novel concept recognition
* The method is model-agnostic and can work with any LMM or text-to-image generative model

Weaknesses:
* As I understood it, the method requires identifying a "confusable concept" for each target concept (one at a time), which could be challenging in some domains where the confusion patterns are not clear.
* Limited discussion on potential failure modes or limitations of synthetic image generation.
* Always valuable to have a dedicated section to future work or potential improvements, problems are never sold and would be useful to other scientists to have more valuable insights on how this work could be carried out further.

**Questions For Authors:**

My questions related to CoDA's sensitivity to various factors, such as:
1. How sensitive is CoDA to the quality of the identified "confusable concept", impact on performance?
2. Performance changes to varying discriminability/generability thresholds?
3. Increasing the number of synthetic images does not necessarily improve performance. Is there a way to predict in advance how many synthetic images would be optimal for a given concept?

**Relation To Broader Scientific Literature:**

The authors position their work well, addressing the existing literature on few-shot image recognition, visual data augmentation, and large multimodal models.

**Theoretical Claims:**

No significant theoretical claims were made that required rigorous mathematical proofs. The paper is focused primarily on empirical and practical improvements to LMMs' visual recognition capabilities.

---

> ### Author Rebuttal · Authors · 2025-04-01
>
> We sincerely appreciate Reviewer pV1R for the insightful review. We are encouraged that you find:
> * **our claims well-supported by evidence**,
> * **our evaluation criteria and benchmarks appropriate**,
> * **our experiments sound and well-executed**.
>
> We are also glad that you **enjoyed reading our submission**, while finding it **well-written, well-structured, with good discussion**.
>
> Please find your suggestions carefully addressed below:
>
>
> ---
> **Discussion on Failure Modes of Synthetic Image Generation**
>
> We appreciate the reviewer for this constructive suggestion. **Given more space in the camera ready version, we will add the following section in the main paper to explain failure modes in synthetic image generation, along with image examples.**
>
> As with any neural generative approach, our method is inherently constrained by the underlying T2I model’s capacity to represent the real world. While we expect this phenomenon to be gradually diminished with the advent of newer and more powerful T2I generative models, as of current publication we can still observe occasional generation results of nonsensical or biologically implausible artifacts (e.g., two-headed snakes) in rare cases. Additionally, while CoDA emphasizes class-discriminative features, it does not explicitly control for other attributes in the generated images. As a result, there can be unintended biases—for example, the backgrounds in generated images for some classes can be quite similar. While we did not observe any significant impact of such biases to our novel and confusing concept recognition tasks, users adapting our method to different use cases should be aware of potential unintentional impacts.
>
> ---
> **Dedicated Future Works Section**
>
> We fully agree with the reviewer that problems are never solved and would be useful to provide more insights on how this work could be improved upon or carried out further. **Given additional space in the camera ready version, we will add the following future works section to our main paper:**
>
> While a main contribution of our work is providing a state-of-the-art visual data augmentation strategy, we leave the downstream innovation on how best to use our augmented visual data to improve models for future work. In our experiments, we focus on the fine-tuning use case as it is the most general and intuitive way to utilize our augmented visual data. Besides this, other conceivable potential use cases for our augmented data include model adaptation, test-time augmentation, visual RAG or even pre-training. The modularity of our method also invites other researchers to replace components of CoDA with superior models to achieve better performance. The NovelSpecies dataset, which will continue to be updated with new species every year, may also be used to evaluate future VLMs' novel concept recognition abilities. Finally, we also expect improved versions of T2I generation-based visual data augmentation techniques to eventually surpass CoDA in effectiveness and efficiency, potential improvements may include more robust image / feature filtering and more controllable text-conditioned image generation like multi-view synthesis.
>
> ---
> **Computational Overhead Comparison**
>
> We thank the reviewer for bringing up this point: While we have thoroughly discussed the computational cost of CoDA in our appendix A.3.(L629-654), it would be very helpful to additionally compare this cost with that of existing visual data augmentation baselines such as ARMADA, so practitioners may reference it when considering adoption. **We plan to provide this information in the camera ready version via an additional figure.**
>
>
> ---
>  **Explanation of "Confusable Concept"**
>
> We are glad that the reviewer brings up this point, which we would like to clarify: While CoDA chooses a confusable concept for each target concept, this process is very general and simply based on model misrecognition.
>
> For example, when LMMs are tasked to classify a novel concept it has no previous knowledge of, the VLM will simply hallucinate and provide the closest confusable concept. CoDA then teaches the VLM the target concept by illustrating the visual differences between the target concept and its corresponding confusable concept. This effectively reduces the difficult and costly task of learning this novel concept from scratch to the much simpler task of learning the difference between the target concept and its confusable concept.
>
> The effective learning cost reduction will depend on the level of similarity between the target concept and its closest confusable concept. However, we believe that as VLMs become more knowledgeable, they will also become better at finding higher quality similar confusable concepts, and thus dramatically reducing the cost of learning to recognize novel concepts.

---

### Decision · Program_Chairs · 2025-05-01

**Decision:**

Accept (poster)

**Comment:**

Reviewers agree this paper addresses a practical and timely challenge in LMMs through a novel contrastive data augmentation framework, CoDA. Reviewers agree that the method is well-motivated, well-executed, and broadly applicable. The authors provide strong empirical evidence across diverse datasets and models, including a valuable new benchmark (NovelSpecies). Rebuttals are detailed and constructive, with thoughtful clarifications and additional experiments addressing reviewers’ concerns, including naive baselines and TTA. While some limitations in scope and compute claims exist, they are revised in the response. Overall, the work is a solid extension of data-efficient adaptation for multimodal models.